# Postharvest Technologies for Fresh Citrus Fruit: Advances and Recent Developments for Loss Reduction during Handling and Storage

**Maria Concetta Strano ¹, Giuseppe Altieri ², Maria Allegra ¹, Giovanni Carlo Di Renzo ², Giuliana Paterna ², Attilio Matera ² and Francesco Genovese ²,***

¹ Centro di Ricerca Olivicoltura, Frutticoltura e Agrumicoltura, Consiglio per la Ricerca in Agricoltura e L'analisi dell'Economia Agraria (CREA), Corso Savoia 190, 95024 Acireale, Italy; mariaconcetta.strano@crea.gov.it (M.C.S.); maria.allegra@crea.gov.it (M.A.)

² Scuola di Scienze Agrarie, Forestali ed Ambientali (SAFE), Università degli Studi della Basilicata, Via dell'Ateneo Lucano 10, 85100 Potenza, Italy; giuseppe.altieri@unibas.it (G.A.); giovanni.direnzo@unibas.it (G.C.D.R.); giuliana.paterna@unibas.it (G.P.); attilio.matera@unibas.it (A.M.)

* Correspondence: francesco.genovese@unibas.it; Tel.: +39-0971205256

**Abstract:** *Citrus* spp. are spread mainly in the Mediterranean basin and represent the largest fruit source for human consumption. Postharvest losses, mainly due to diseases and metabolic disorders of fruits, can cause severe wastage, reaching 30 to 50% of the total production. Preserving quality and extending shelf life are essential objectives for postharvest technological innovation, determined by the proper handling, treatment, storage and transport of harvested produce. Moreover, the application of novel sustainable strategies is critical for the reduction of synthetic fungicide residues on fruit surfaces and the impact on the environment caused by waste disposal of fungicides. In this article, the current knowledge about the safest and more sustainable strategies, as well as advanced postharvest handling and storage technologies, will be critically reviewed.

**Keywords:** sustainable strategies; innovative technologies; fruit quality; storage; shelf life; postharvest; modified atmosphere packaging (MAP); cold storage room; precooling

## 1. Introduction

*Citrus* spp. are the largest fruit source for human consumption in more than 100 countries distributed over the world. Citrus crops are grown chiefly in the Mediterranean Basin, particularly in Spain, Greece, Italy, Tunisia and Turkey, as well as Australia, California and South Africa—countries and regions characterized by Mediterranean-like climates [1]. Oranges (*Citrus sinensis* (L.) Osbeck), mandarins (*C. reticulata* L.), clementines (*C. clementina* Hort. ex Tan.), lemons (*C. limon* (L.) Burm.), grapefruits (*C. paradisi* Macf.), limes (*C. aurantifolia* Swing), pummelos (*C. maxima* (Burm.) Merrill), kumquats (*C. japonica* Thumb.) and citrons (*C. medica* L.) are the species mainly cultivated throughout the world [2]. Tarocco orange is a pigmented variety traditionally grown in the Mediterranean basin, but also spread across other areas, such as China, Australia, and the United States. In Italy, Tarocco is almost entirely cultivated on the Eastern coast of Sicily, in the foothills of the Etna volcano [3].

Citrus fruits are particularly appreciated for their positive health benefits, representing an important source of bioactive compounds with high antioxidant activity, such as vitamin C, hydroxycinnamic acids and flavonoids [4]. Postharvest losses, mainly due to diseases (rots) and physiological disorders, can cause severe waste of fruits, ranging between 30 and 50% of the total production after 30 to 60 days of storage [5]. Postharvest technology for fresh citrus fruit encompasses several techniques and

strategies studied and applied to improve the different steps of handling, processing, storage, and transport of fruit, which still suffer from critical aspects. The strategies are elaborated in each citrus-producing country based on their requirements, target destinations and other factors, such as local law restrictions [6].

Thus, preserving the quality of freshly harvested fruit, extending their shelf life and reducing fruit losses are the most important goals of postharvest technological progress.

In this review, research papers published over the past six years concerning the postharvest management of citrus, starting with diseases and physiological disorders and proceeding to the safest sustainable strategies and advanced postharvest handling, storage and analysis technologies, will be critically examined. This review aims to integrate existing information from other authors' reviews [6–8] with the latest news concerning the early detection of citrus fruit alterations during handling and storage. The internet search engines used were Scopus (https://www.scopus.com, accessed on 24 June 2022), PubMed (https://www.pubmed.ncbi.nlm.nih.gov, accessed on 24 June 2022), Web of Science (http://apps.webofknowledge.com, accessed on 18 June 2022) and Google Scholar (https://scholar.google.com, accessed on 20 June 2022).

The review is organized into three main sections dealing with the following topics: (i) postharvest diseases and physiological disorders of citrus fruit; (ii) novel technologies and strategies for the control of fruit alterations; (iii) advanced postharvest technologies for the management of fresh citrus fruit quality during handling and storage.

## 2. Causes of Postharvest Loss in Citrus Fruit

The primary causes of postharvest citrus fruit losses can be split into two major groups: (i) physical loss due to diseases and injuries and (ii) quality loss due to metabolic and compositional changes in the fruit. However, peel injuries and impact bruising are the leading causes of fruit spoilage, as they promote pathogen growth, respiration, ethylene production and significant water loss, with the consequent wilting of the product [9].

### 2.1. Postharvest Diseases

Rots caused by fungal pathogens are considered the most significant cause of citrus fruit waste and quality deterioration as they make fresh fruit unfit for consumption, resulting in severe economic losses. They can be distinguished by the time of infection: some are caused by infections of unripe fruit in the orchard, which show symptoms after a period of dormancy, while others are caused by infections that occur at harvest and during the postharvest phase [10].

Preharvest infections include Alternaria rot (*Alternaria citri* Ellis et Pierce.) and Brown rot (*Phytophthora* spp.). Although they have a low incidence, these infections can be a severe issue in warm and wet years [11].

The most frequent and severe postharvest diseases which affect citrus yield are green mould and blue mould, caused, respectively, by the wound pathogens *Penicillium digitatum* Sacc. and *P. italicum* [12]. Fungal spores, produced massively by rotten fruit, propagate through the air and can easily contaminate adjacent fruit at any stage. *P. digitatum* can cause 60–80% decay under ambient conditions, while *P. italicum* is observed chiefly on cold stored fruits [13]. Other important postharvest diseases are sour rot (*Galactomyces citri-aurantii* E.E. Butler) and gray mould (*Botrytis cinerea* Pers.: Fr.). Infrequent postharvest pathogens are *Penicillium ulaiense*, (Whisker mold), *Trichoderma viride* Pers. (*Trichoderma* rot), *Aspergillus niger* Tiegh (*Aspergillus* black rot) and *Rhizopus stolonifer* (Ehrenb.) Vuill. (*Rhizopus* rot).

*Phomopsis citri* Fawcett, *Lasiodiplodia theobromae* (Pat.) Griffon and Maubl. (syn.: *Diplodia natalensis* Pole-Evans) and *Diaporthe citri* Fawc., which cause stem-end rot, are considered latent pathogens, as well as *Colletotrichum gloeosporioides* (Penz.) Penz. & Sacc., the cause of anthracnose. These pathogens infect fruit in the orchard during the growing season and remain quiescent until they start growing again after harvest [11].

Most of these rots often go undetected during sorting in packing lines, especially in the early stages of infection of the fruit, so they spread later, propagating to adjacent healthy fruit and forming mycelium nests, consequently deteriorating entire boxes or bins of produce [6].

*2.2. Physiological Disorders*

Citrus fruits are not climacteric; therefore, under normal conditions, they are not affected by rapid changes in respiration and ethylene production, as are climacteric fruits. Respiration rate decreases slowly until senescence, whereas ethylene production is deficient (0.1 mL kg$^{-1}$ h$^{-1}$ at 20 °C) and does not show any peak [14]. Sudden metabolic changes, such as increased respiration, higher ethylene production and accelerated transpiration rate, which occur due to early fruit senescence or stress resulting from inadequate postharvest handling and storage, are precursors of physiological disorders [15].

The primary commercially relevant physiological disorders, originating due to inappropriate storage temperatures, are chilling injury and membranosis. Nevertheless, various non-chilling peel disorders can occur at non-chilling temperatures due to incorrect fruit handling after harvest, which may negatively impact the quality of fruit, resulting in different types of blemishes. These include oleocellosis, stem-end rind breakdown, peel pitting and peteca in lemons [16].

Chilling injury (CI) is a severe physiological disorder that occurs in fruit held in a thermal range above freezing point (0–8 °C) for an extended period. However, the susceptibility and the extent of the damage vary widely among the different citrus species and cultivars. CI affects the rind with the development of tiny to large collapsed brown areas over the surface of the fruit, compromising its external quality and marketability. As a consequence of CI, it is possible to observe an increase in mould development on fruit surfaces [16]. Although symptoms develop during cold storage, often they become more noticeable only after the fruit is brought back to warmer temperatures, such as during marketing. Lemons, grapefruits and limes are more sensitive to CI, followed by oranges, mainly pigmented ones, and mandarins [17]. Considering the commercial importance of citrus fruit, their responses to postharvest cold stress have probably been the most studied mechanisms over the years. Several authors [16,18–20] have indicated the damage to cell membranes as the primary event in CI development. As a result of membrane disorganization, various metabolic processes are involved, causing ion leakage and the overproduction of reactive oxygen species (ROS), with subsequent cell death and, finally, the pitting of surfaces. Molecular studies have found that CI switches on the expression of genes related to membrane lipid and sterol metabolism, thus indicating a key role for lipid metabolism in the response of citrus fruits to cold [21].

Membranosis is a lemon-specific CI that can reach an incidence of up to 90% during cold storage. It develops on the inner membranes as reddish-brown and dark areas in the albedo [15].

Oleocellosis is a disorder usually produced by rough fruit handling during harvest and processing. It causes the rupture of oil glands and leakage of their contents, producing a phytotoxic effect and rind cell necrosis [16].

Stem-end rind breakdown (SERB) occurs with an area of brown necrotic rind around the calyx end of the fruit caused by high fruit transpiration. Differences in the number of stomata and the thickness of the wax cuticle surrounding the calyx may explain the peculiar site of the disorder. Generally, fruits with thinner rinds are more susceptible to SERB [22].

Peel pitting or staining indicates a skin alteration that occurs at non-chilling temperatures several days after the fruits have been processed in the packing line. This disorder is caused by the collapse of the oil glands in the fruit peel, which then turn dark brown. In Florida, the disorder was initially observed in wax-coated grapefruits, so it was hypothesized that altered internal gas concentrations, caused by less permeable shellac-based waxes, were responsible [23]. The disorder was later induced in non-waxed fruit,

showing that the waxing itself was not the direct cause of the skin pitting [16]. Fruit wax increased the severity of the damage only if there was a previous period of dehydration [24].

Postharvest physiological disorders can generally be reduced by avoiding harvesting fruit in wet conditions, e.g., after rains, cooling citrus fruit immediately after harvest and storing it at appropriate temperatures and high relative humidity, as well as considering species and varieties, as shown in Table 1.

**Table 1.** Recommended temperature ranges, relative humidity (RH) values and approximate storage lives for citrus fruit species.

| Citrus Species and Variety | Temperature Range (°C) | RH % | Approximate Storage Life |
|---|---|---|---|
| Grapefruit | 10–15 | 85–90 | 6–8 weeks |
| Lemon | 10–13 | 85–90 | 1–6 months |
| Limes | 9–10 | 85–90 | 6–8 weeks |
| Mandarin hybrids (Fortune, Nova) | 8–9 | 85–90 | 4–6 weeks |
| Mandarin, Tangelo | 5–6 | 90–95 | 2–4 weeks |
| Clementine, Satsuma | 4–5 | 90–95 | 2–4 weeks |
| Kumquat | 4–5 | 90–95 | 2–4 weeks |
| Pigmented orange | 6–8 | 90–95 | 3–8 weeks |
| Blond orange | 3–9 | 85–90 | 3–8 weeks |

Adapted from: Strano et al. [4]; FAO [25]; USDA [26].

## 3. Novel Citrus Postharvest Technologies and Strategies

Current postharvest decay control strategies rely primarily on synthetic chemical fungicides to maintain the quality of citrus fruit after harvest. Imazalil is the fungicide most used for postharvest treatment of citrus fruits [27]. Consumer demand for fruit free of chemical residues, the risk of environmental impact, the occurrence of resistant pathogen populations and increasing legislative restrictions have urged researchers to explore and develop novel, safer and more environmentally friendly strategies [28].

In recent decades, several studies have been carried out on citrus fruit to evaluate the effect of physical treatments, emerging chemical strategies, biocontrol agents and edible coatings in controlling *P. digitatum* (green mould) and *P. italicum* (blue mould) development [29].

### 3.1. Physical Treatments

3.1.1. Heat Treatments

Heat treatments (HTs) are physical means of great interest as they leave no residues, are relatively effective in controlling postharvest decay and improving fruit disease resistance, are easy to apply and combine, and are synergistic with other control methods [6]. HTs can be applied to citrus fruit using forced hot air and hot water. However, depending on the temperature and exposure time of the treatment, fruit quality parameters risk being compromised [30–38]. Hot air treatment or curing, hot water systems (hot water dips (HWDs) and hot water rinsing and brushing (HWRB)), have been widely adopted in several countries to inhibit rot development in different species and cultivars of citrus fruit and to alleviate chilling injury in the postharvest phase [30–33,36–41].

Curing consists of exposure of citrus fruit to a heated air temperature between 37–45 °C and relative humidity (RH) > 90% for 2–3 days [30,36–38].

Preconditioning treatment at 37 °C for 48 h effectively reduced decay incidence, weight loss and respiration rate in navel oranges stored at 6 ± 0.5 °C and 85–90% relative humidity (RH) for 120 days [37]. Moreover, the enhanced enzyme activity of superoxide

dismutase (SOD), peroxidase (POD) and polyphenol oxidase (PPO) was also observed in fruits stored at low temperature, highlighting a potential effect of curing in improving tissues' resistance to diseases [37]. However, although it has proven effective, especially against *Penicillium* moulds, the commercial application of curing is rare because it appears to be impractical due to the heating expense and the difficulties in immobilizing large amounts of fruit. Moreover, it can cause high fruit weight loss or rind damage if not correctly applied [32,34].

HWD is generally applied as relatively brief immersion (1–5 min) of fruit in water heated at 40–55 °C, whereas HWRB basically involves packing line machinery that applies hot water over rotating brushes at high temperature (55–65 °C) for a short time (10–30 s) [39]. Valencia oranges (*Citrus sinensis* (L.) Osbeck) dipped in water at 53 °C for 3 min and subsequently stored at 4 °C (RH 90–92%) for six months showed a significant reduction in fruit decay incidence (4.80%) with respect to the control (24.09%) at the end of storage [28]. HWRB treatments at 56 °C for 20 s reduced decay in organically grown tangerines, oranges and red grapefruits, without affecting fruit quality [40].

HWD and HWRB are more practicable than curing but still too expensive for commercial application on citrus fruit and less effective and persistent; moreover, the range of effective but non-phytotoxic temperatures is very narrow. Too high temperatures and the prolonged duration of treatment can cause tissue damage in sensitive species; thus, treatments at optimum temperatures and with optimum time intervals are recommended for each cultivar [30]. Lanza et al. [31] evaluated the efficacy of HWD (52 ± 1 °C for 3 min) and HWRB treatments (62 ± 1 °C for 20 s) in reducing green mould (*P. digitatum*) in Tarocco and Valencia oranges (*Citrus sinensis* (L.) Osbeck) and 'Femminello siracusano' lemons (*C. limon* (L.) Burm). The results showed the significant efficacy of HWD on Tarocco oranges (0%) compared with HWRB (40%) and controls (76%). In lemon fruit, both treatments reduced green mould incidence (20%) compared to controls (99%).

In current applications, HWD is limited to some organically grown citrus fruit [32]. In contrast, pre-storage HWD could be considered an efficient method suitable for commercial application to prevent decay and chilling damage during storage and improve the storage stability of different citrus fruit species [39,41,42]. Hong et al. [33] reported the efficacy of HWD at 60 °C for 20 s in preserving the qualitative traits of an early harvesting cultivar of Satsuma mandarins (*Citrus unshiu* Marc., cv. Gungchun) and improving fruit appearance after three weeks of storage at 5 °C.

Despite their limitations, heat treatments play a key role in the environmentally friendly integrated strategy for postharvest decay control [6,41].

### 3.1.2. Irradiation

Irradiation is a non-thermal safe technology approved by the World Health Organization (WHO), the Food and Agriculture Organization (FAO), the International Atomic Energy Agency (IAEA) and the Codex Alimentarius Commission for the postharvest treatment of fresh fruit and vegetables [43]. It can be applied by ionizing radiations, such as gamma rays ($^{60}$Co or $^{137}$Cs), electron beams (β particles) and X-rays, or by non-ionizing radiations, such as ultraviolet light, both in bulk and packaged food products [44]. Application of irradiations at a low dose (<2 kGy) delays sprouting of vegetables and ageing of fruits; at a medium dose (1–10 kGy), it deactivates pathogen growth; and at a high dose (>10 kGy), it sterilizes the treated produce. Dosages of 50–750 Gy are required for insect pest control, whereas higher doses (1000–1750 Gy) are required for postharvest disease control [45].

Several authors have shown the potential of the application of non-ionizing (UV-C, UV-B, blue light) and ionizing irradiations (gamma and X-rays) in reducing the incidence of fungal diseases in citrus fruit [46–56].

Kinnow citrus fruits treated with a radiation dose of 1.5 kGy showed a reduction in mould incidence [57]; combined treatments with gamma irradiation (doses up to 1.5 kGy) followed by refrigerated storage were effective in delaying *Penicillium* rot in 'Nagpur'

mandarins, 'Mosambi' sweet oranges and 'Kagzi' acid limes [58]. Oufedjikh et al. [59] showed the enhanced synthesis of total phenolic compounds while storing Moroccan *Citrus* fruits. Gamma-ray irradiation less than 1 kGy showed potential as a postharvest phytosanitary irradiation procedure for eliminating quarantine pest treatment in Korean citrus fruits [51].

Electron beam (eBeam) technology has recently been investigated as an alternative to thermal postharvest treatments for fresh produce for phytosanitary purposes [52]. With an eBeam linear accelerator, generated electrons are accelerated to very high velocities (about 99.999% of the speed of light), gaining energies of up to 10 MeV (million electron volts) [53]. These electrons can then enter rapidly into the product, causing double-strand breaks in DNA and thus the inactivation of microorganisms present on the target product. Regarding applications in the food industry, high energies (5–10 MeV) are required for phytosanitary treatments [53]. The energy (dose rate) applied to the target products is measured in kilograys (kGy). The effective dose needed to deactivate pathogens is 1–10 kGy; however, the maximum eBeam dosage allowed in the U.S. by the FDA for the processing of fresh produce is 1 kGy [54].

Recent studies carried out by Nam et al. [60] on mandarins (*Citrus unshiu* (Swingle) Marcov) during storage at 4 °C for 15 days demonstrated that eBeam irradiation at dosages up to 0.4 kGy did not affect the major constituents and physical quality of mandarins and avoided microbial proliferation (total aerobic bacteria, yeasts and moulds, and coliform counts). Ramakrishnan et al. [61] reported that eBeam doses of 1 kGy minimize quality deterioration in grapefruit and lemons stored at 4 °C for 20 days.

Irradiation with X-ray doses of 510 and 875 Gy of *Penicillium*-infected 'Clemenules' mandarins resulted in the inhibition of pathogen sporulation without inducing negative effects on fruit quality [62]. Alonso et al. [63] indicated X-ray irradiation as a harmless and highly effective quarantine technique for clementine mandarin cv. 'Clemenules', potentially as useful as the current cold treatment for this type of mandarins. However, the current results do not justify X-ray irradiation use under commercial conditions.

Ultraviolet radiation (UV; 100–400 nm), used to control rot development, has also proven effective in delaying fruit senescence and increasing the production of beneficial compounds [46]. UV-C light is the most widely used frequency range (200–280 nm) on account of its germicidal properties [55] and for its ability to induce defence response mechanisms [64] against most microorganisms, including fungi, yeasts, bacteria, viruses, protozoa and algae. In cold-stored citrus fruits, induced resistance was linked to the build-up of phytoalexins, such as scopoletin and scoparone [56]. Phonyiam et al. [65] demonstrated that application of UV-C at 10 kJ m⁻² on Satsuma mandarins was able to reduce the growth of *P. digitatum* (green mould), maintaining the integrity of the membrane structure by reducing lipid peroxidation and increasing jasmonic acid accumulation with the induction of bioactive compounds and the antioxidant potential of DPPH radical scavenging capacity. Yamaga et al. [50] showed that low UV-B doses (15 kJ m⁻²) had an inhibitory effect on *P. italicum* growth in vitro, with a reduction in spore germination >99%. In any case, due to their low penetration capacities with respect to fruit surfaces, the irradiation techniques must be applied by integrating alternative strategies. Integrated applications, with GRAS compounds, heat treatments, cold storage and antagonistic microorganisms, remain efficient approaches, since their synergistic effects result in improved effectiveness and better results [62].

### 3.1.3. LED Blue Light

LED (light-emitting diode) blue light treatments (LBL) have recently been investigated and proposed as eco-friendly, safe and low-energy cost technologies, beneficial for pathogen growth control [66–68] and able to improve citrus crop production and nutritional quality. Lafuente et al. [68] reported that LBL treatment at a quantum flux of 60 μmol m⁻² s⁻¹ for 2 days reduced decay caused by *P. digitatum* in Lane Late oranges. Ballester and Lafuente [69] demonstrated that blue light at 450 nm, 630 μmol m⁻² s⁻¹

induced some phenylpropanoids, such as scoparone. Other studies reported the results of the development of elicit resistance against *P. digitatum* in citrus fruit treated with LBL [70,71], as well as the accumulation of carotenoids [72] and ascorbic acid [73].

Despite the encouraging results obtained, the treatment duration of LBL needed for effective decay control remains an obstacle to its practical application. In order to reduce the time duration of LBL treatment, the effect of an increased LBL quantum flux (630 μmol $m^{-2}$ $s^{-1}$) on fungal growth in inoculating oranges was assessed by the evaluation of increased elicited resistance. Ballester and Lafuente's [69] results, obtained after 18 h of treatment, confirmed that the highest quantum flux was able to induce resistance against *P. digitatum* in only 3 h. However, the efficacy of this treatment was poor for effective decay control. Thus, further studies are needed to improve LBL treatment efficacy by integrating it with other strategies.

### 3.1.4. Other Emerging Non-Thermal Technologies

Pulsed light (PL) and power ultrasound (PU) are novel, non-thermal promising strategies that have been studied over the past few years for food preservation.

Pulsed light (PL) is designed to extend the shelf life of food through the microbial decontamination of surfaces, such as food packaging material and equipment, and to preserve or improve the nutritional and sensory quality of food [74]. It is based on a series of extremely short-duration (1 μs–0.1 s), high-frequency (0.01–50 $J/cm^2$) pulses generated by an electronic oscillator discharging into xenon or xenon–mercury lamps. The U.S. Food and Drug Administration (FDA) has recently approved PL technology for food treatments at maximum radiant energy (Fluence) of 12 J $cm^{-2}$ [75]. Recent investigations have evaluated PL efficacy in pathogen inactivation on the surface of fresh products [76,77], with promising results in controlling pathogen growth in various fruits [78,79].

Power ultrasound (PU) is used for food decontamination [80,81] and to preserve the quality of fresh fruits and vegetables, cereal and bread products, commercial cooking oils and food gels [82,83]. PU refers to the section of the sound spectrum from 20 kHz up to around 1 MHz. The foundation of many ultrasound treatments in this frequency range is acoustic cavitation, which is the generation, expansion and breakdown of microbubbles in an aqueous solution that cause increases in localized pressure and temperature in a transmission liquid medium, damaging cell walls and membranes and causing enzyme denaturation of fresh food [84–86].

Although no studies on citrus fruit have yet been carried out, both technologies can be considered new approaches in the fruit industry that avoid or reduce the use of chemical fungicides, with broad potential application prospects in preserving bioactive substances and physicochemical properties and improving the quality and storage life of fresh fruit [86].

### 3.1.5. Cold Atmospheric Plasma (CAP)

Cold atmospheric plasma (CAP) is a non-thermal emerging technology that, over the last few years, has found many applications in the food industry. This process is usually carried out at room temperature, under atmospheric pressure or vacuum, for microbial decontamination and enzyme inactivation of food [87,88].

Plasma generation can be obtained through different electrical discharge methods, such as corona discharge, dielectric barrier discharge, glow discharge, high-frequency discharge or microwaves, using carrier gas air, nitrogen, oxygen, noble gases or different gas mixtures [88]. Consequently, reactive species, such as photons, free electrons, charged molecules and free radicals, are generated. The antimicrobial effectiveness of CAP is based on the production of reactive oxygen and nitrogen species (RONS), such as $O_3$, $O_2^-$, $OH^-$, NO and $H_2O_2$, and UV radiation, known for its broad activity against bacteria and fungi, which cause erosion of cell surfaces and cell membrane disruption [89]. The combination of air with argon or sulphur hexafluoride has also been recently studied for postharvest disease control in fruit and vegetables [90].

Several authors [88,91–94] have reported the efficacy of CAP processing in microbial control of different fresh fruit and vegetables and the absence of detrimental effects on physicochemical and sensory properties. Recent developments of this technique include the application of CAP within packaging to extend the shelf life of fresh-cut fruit and vegetables, and there is great interest in its commercial applications due to its lack of toxicity and minimal production of residues [95,96].

Regarding citrus fruit, Won et al. [97] reported that the viability of *Penicillium italicum* was notably inhibited (84%) on artificially inoculated Satsuma mandarin peel (*Citrus unshiu* Marc.) treated with a cold microwave plasma (CP) system, using nitrogen gas at 900 W for 10 min at a pressure of 0.7 kPa and a gas flow rate of 1000 mL·min$^{-1}$. The same authors showed that the same CP conditions applied to whole Satsuma mandarins stored in Whirl-Pak® bags (Nasco, Fort Atkinson, WI, USA) for 35 days at 4 ± 1 °C did not alter the fruits' biological properties or quality and instead increased the total phenolic content and antioxidant activity of the peel. Puligandla et al. [91] observed that kumquats treated using an intermittent corona discharge plasma jet (output voltage: 8 kV DC; current levels: 2.0–4.0 A) exhibited improved microbiological quality and extended shelf life. In a recent study, Sakudo and Yagyu [98] reported a decrease in viable cell number of *Penicillium venetum* on different treated Kiyomi (*Citrus unshiu* × *C. sinensis*), Kawano-natsudaidai (*C. natsudaidai*) and Unshu-mikan (*C. unshiu*) fruits, using a novel roller conveyer plasma device in which plasma is generated by an atmospheric pressure dielectric barrier discharge. Treatments were carried out connecting the high-voltage electrode to an alternating power supply (VPP peak-to-peak voltage = 11.87 kV; frequency = 8.85 kHz). The estimated treatment time required to reduce *P. venetum* viable cell number by 90% (D value) on the fruit surface was calculated. The following values were obtained: 2.90 min for Kiyomi, 1.88 min for Kawano-natsudaidai and 2.42 min for Unshu-mikan.

### 3.1.6. Precooling

Rapid cooling is a consolidated technology that is able to quickly lower the temperature of freshly harvested fruits and vegetables to reach the thermal level applicable during storage, cold treatment [99] and shipment to market, resulting in substantial reductions of both weight loss and decay. Generally, rapid cooling/precooling is defined as the time needed to lower the "field heat" and the time required to remove 7/8 of the field. Different systems for rapid cooling are nowadays available, especially in industrialized countries, such as room cooling (by chilled air), forced air cooling (FAC), hydrocooling (by chilled water) and vacuum cooling [5].

Due to easy management and for economic reasons, the 'room cooling' system has been widely used for precooling. In the room cooling system, the heat exchange between fruits (packed in cartons, sacks or bins) and cold air takes place directly in cold storage rooms using air fans blowing cold air (air speed between 1 and 2 m s$^{-1}$). Once the final temperature has been reached, the air velocity is reduced to about 0.05–0.1 m s$^{-1}$.

Nowadays, the use of traditional cold storage rooms has been overtaken as it is considered adequate only for the storage of fruits already chilled because the mechanism of cooling is too slow and is not suitable for the quick removal of 'fruit heat' (the thermal level of fruit when harvested in the field). Moreover, it leads to both inefficient (prolonged) cooling and excessive water loss (associated with over-drying of citrus skin) [4]. Therefore, precooling is essential to reduce the physiological response to postharvest stress and enables reductions in disorder incidence during storage.

Among the various types of precooling systems available nowadays, FAC represents the most effective method for citrus fruit [100]. It is used in combination with a cooling room consisting of fans and strategically placed barriers so that cold air is forced to pass through the fruits, stacked in pallets, with a remarkable reduction in the time required to cool the product by passive room cooling. Several authors [101], considering citrus fruits stacked in pallets, investigated cooling performance in terms of cooling rate, convective

heat transfer coefficient and energy consumption, and also evaluated the impact of packaging design and fruit size on cooling rates [100].

Recently, Elansari and Mustafa [102] studied the effect of the cooling rate of the Navel orange as a function of fruit size, air direction (vertical forced and vertical induced) and air velocity on weight loss and electrolyte leakage, with specific reference to Citrus packinghouses in Egypt. The authors concluded that, despite the fast cooling of fruits being generally ignored by many Egyptian citrus packinghouses, as well as in other countries, it is a severe concern for different markets in the light of improving citrus fruit quality.

In another study [13], the authors, considering the extension of fruit shelf life an excellent means of increasing the incomes of growers of Clementine mandarins in Southern Italy, evaluated the effectiveness of precooling combined with both HDPE wrapping packaging and a patented gas exchange device (BlowDevice®, BlowDevice Ltd., Potenza, Italy) in terms of fruit weight loss reduction and clementine quality increase. In the work [13], the authors concluded that forced-air cool treatments of packaged clementines reduced weight loss and physiological disorders and showed acceptable fruit quality. Within 30 days of cold storage, weight loss was limited to 1–2% for precooled and wrapped fruits compared to 5–8% for unwrapped fruits. In addition, reasonable control of physiological disorders and maintenance of quality parameters was obtained, even after one week of shelf life.

### 3.1.7. Modified and Controlled Atmosphere Storage

Modified atmosphere packaging (MAP), using different types of packaging materials, such as Xtend® (StePac L.A. Ltd., San Diego, CA, USA), polyethylene (PE) and polyvinyl chloride (PVC) films, may be a possible means of reducing the development of postharvest rind disorders in citrus fruits, including SERB and chilling injuries. As reported by several authors, MAP acts by replacing the surrounding air atmosphere concentration around the fruit inside a sealed package with the $O_2$ and $CO_2$ levels required to mitigate pathological decay and physiological disorders [103–107]. Creating low $O_2$ and medium–low $CO_2$ partial pressures, which can reduce respiration rate and fungal growth, and maintaining a high RH, which reduces water loss and therefore weight loss, MAP may also be used as a commercially practical means to extend the shelf life of citrus fruits, retaining freshness and preserving initial fruit quality during storage [108–110]. However, supposing that the permeability of the films used is not appropriate, MAP can promote anaerobic respiration and the accumulation of excessive moisture inside the package, particularly at non-chilling temperatures, with detrimental effects, such as the development of off-flavours or increase in the incidence of decay [111,112]. To control pathological decay without causing physiological damage or injuries, it is critical to select films with a low moisture transmission rate and selective permeability to $O_2$ and $CO_2$, sometimes in combination with the use of chemical or other postharvest treatments, such as heat treatments, irradiation or natural antifungal compounds [112].

In MAP applications, the initially modified $O_2$ and $CO_2$ concentrations change during storage, depending on film permeability and fruit respiration rates and temperature, while in controlled atmosphere (CA) storage, the concentrations of $O_2$ and $CO_2$, as well as the temperature and humidity, are continuously monitored and adjusted by special equipment designed to achieve exact control of gases [113]. In general, CA storage is a supplementary technology to refrigerated storage to extend the shelf life of fruits, but better understanding is needed for its exploitation in citrus fruit production. Several attempts have been made to preserve the main quality indexes related to flavour, odour and nutritional profile using CA.

By means of CA storage, using high $CO_2$ concentrations, fungal decay is controlled, this being required to control most citrus fruit postharvest pathogens; furthermore, it is not tolerated by fruit, resulting in the development of rind disorders or off-flavours

related to the accumulation and threshold levels of the products of anaerobic respiration [113,114].

Low $O_2$ or high $CO_2$ atmospheres stimulated the accumulation of ethanol, acetaldehyde and several other volatile compounds in oranges [115–117] and increased the activity of pyruvic decarboxylase and alcohol dehydrogenase [115].

Prestorage treatments from 3 to 7 days with 10% to 40% $CO_2$ reduced chilling injury and stem-end rind breakdown of grapefruits [118].

Ke and Kader [119] investigated the effects of short-term exposures (20 days) to $O_2$ levels at or below 0.5% and a $CO_2$ level of 60% at several temperatures on postharvest physiology and quality attributes of Valencia oranges to determine their tolerance to these treatments, which may be used for postharvest insect control, and observed significant increases in ethanol and acetaldehyde contents compared with the control in air, which correlated with a decrease in flavour score of the fruits.

For lemons, limes and Satzuma mandarins, within specific ranges (Table 2), CA can be exploited to mitigate disorders and delay skin degreening [120–123].

**Table 2.** Controlled atmosphere (CA) requirements for some citrus fruits.

| Species | Temperature Range (°C) | Relative Humidity (%) | CA | |
|---|---|---|---|---|
| | | | % $O_2$ | % $CO_2$ |
| Lemon | 10-15 | 90-95 | 5-10 | 0-10 |
| Lime | 10-15 | 90-95 | 5-10 | 0-10 |
| Orange | 5-10 | 90-95 | 5-10 | 0-5 |

Adapted from: Erkan and Wang [120].

Super atmospheric oxygen storage (SAO; at $O_2$ concentrations greater than 21 kPa) is an emerging technology that may influence postharvest physiology and quality maintenance of fresh horticultural perishables either directly or indirectly via altering $CO_2$ and ethylene production rates—depending on the commodity, maturity and ripeness stage—and inhibiting the growth of some bacteria and fungi [124,125].

A recent study carried out on 'Sanguinello Comune' blood oranges stored at 10 °C with 76 kPa of $O_2$ suggested a remarkable enhancement of total anthocyanin accumulation in the fruit juice (over 100 times) accompanied by a significant increase in total phenolic compounds that could be related to the protective response of the fruit toward oxidative stress caused by the oxygen-enriched atmosphere [126]. At the same time, SAO caused a significant decline in acidity, the total content of soluble solids and contents of sucrose, glucose, fructose and ascorbic acid. On the contrary, honey pomelo (*Citrus grandis* L.) slices stored under SAO showed significantly depleted ascorbic acid content and antioxidant capacity without undergoing firmness reduction [127].

In conclusion, the commercial application of controlled and modified atmosphere technologies to citrus fruit should take into account the different tolerances of citrus species and cultivars to specific $O_2$ and $CO_2$ levels, as these treatments can stimulate physiological stress and the accumulation of ethanol, acetaldehyde and other volatile components that lead to off-flavour development [128].

### 3.1.8. Innovative Packaging

Appropriate packaging may be beneficial in maintaining the quality of citrus fruits, reducing the mechanical damage to fruits during transportation and storage and increasing the storage life and therefore the commercial value of products.

Depending on the subsequent mode of transport and storage, citrus fruits are usually packaged in corrugated fiberboard (CFB) boxes, cardboard trays in flow-pack film, net bags or film bags [129].

In the citrus industry, package design optimization has been the subject of much research due to its importance in reducing the forced-convective precooling time and improving citrus fruit quality by providing the fastest and most uniform cooling without chilling-induced damage. Furthermore, packaging should combine the best possible ventilation with mechanical strength, which is required for the stacking of containers and the protection of fruit. Defraeye et al. [130] examined the cooling performance of two new packages, "Supervent CFC" and "Ecopack reusable plastic container" (RPC), stacked on a pallet, which were superior to the existing corrugated fiberboard container (CFC).

Zheng et al. [131] proposed a new packaging format which uses plastic dividers to prevent oranges from crushing each other, with an EPE foam layer in each divider and a PU foam layer placed along the inside of the EPE foam layer to suit the different sizes of fruits. The study results showed that the proposed packaging was more suitable than the existing packaging for road transportation of Hongmeiren orange fruit, a citrus cultivar susceptible to mechanical damage. Although the new packaging uses more materials compared to existing packaging and therefore may cause some environmental problems, it can reduce the waste of resources, such as fertilizers and vehicle oil, resulting from the inability to deliver the fruit to the consumer, as well as environmental pollution caused by fruit decay.

Antimicrobial active food packaging is an emerging technology that allows for the extension of the shelf life of food products through a controlled release of encapsulated antimicrobial compounds [132]. Corrugated cardboard is an eco-sustainable packaging material, widely used for citrus fruits in different formats, such as boxes and trays. Therefore, inclusion complexes with essential oils (EOs) extracted from plants and encapsulated in cyclodextrin (CD) may be included to develop antimicrobial active cardboard packaging and to extend the shelf life of fruits and vegetables [133,134].

López-Gómez et al. [135] studied the effect of active cardboard packaging, including a complex with EOs and cyclodextrin, on mandarin quality stability. The study showed that the controlled release of EOs from the active cardboard box extended the shelf life of mandarins from two to three weeks, even at non-recommended room temperature (8 °C). In Table 3 are listed summary of some studies and their results concerning the application of physical treatments to the control of post-harvest rots.

**Table 3.** Summary of studies on physical treatments effective for postharvest rot control of citrus fruit.

| Physical Treatment | Target Pathogen | Significant Results | Reference |
|---|---|---|---|
| Heat treatments<br>- Curing<br>- Hot water treatments | Postharvest pathogens, especially *P. digitatum* and *P.italicum* | Reduction of decay and chilling injury; induction of disease resistance<br>Curing is impractical under commercial conditions because expensive | [30,32,39,41,42] |
| Irradiations<br>- Ultraviolet radiation | *P. digitatum* and *P.italicum* | Effective in delaying fruit senescence and increasing the production of beneficial compounds | [46,55,56,64,65] |
| Precooling<br>- Forced air cooling | Postharvest pathogens | Improvement of citrus fruit quality by reduction of weight loss and physiological disorders | [13,100] |
| Modified and controlled atmosphere storage | Postharvest pathogens | Retention of freshness and initial fruit quality during storage | [106–112] |

| Innovative packaging | Postharvest pathogens | Preservation of fruit quality, reduction of mechanical damage to fruit during transport and storage, shelf-life increase | [131,132,135] |
|---|---|---|---|

### 3.2. Emerging Chemical Strategies

3.2.1. Sanitizing Agents

Chlorinated water has been used for decades as a sanitizing agent for washing fresh produce to manage postharvest disinfection of fruit and vegetables.

Chlorine dioxide ($ClO_2$) is a strong oxidizing agent, recently proposed as an alternative to sodium hypochlorite—the agent most commonly used—for surface disinfection of fruits and vegetables, thanks to its oxidation capacity being more than 2.5 times greater, its wide biocidal efficacy [136,137] and the lower generation of carcinogenic by-products [138]. Recent studies [139] have investigated the efficacy of $ClO_2$ in reducing the mycelial growth of *Lasiodiplodia theobromae*, the causal agent of postharvest stem-end rot in citrus, in in vitro and inoculated mandarin trials. Complete mycelial growth was obtained after 24 h exposure to solid $ClO_2$, and a higher efficacy was reported for curative assessments, with a decay incidence of 17.6% compared to 95.6% incidence in the control. Gaseous $ClO_2$ has been proved to be more effective in deactivating microorganisms than the aqueous form, as the gas has significant diffusion in cells. However, under practical working conditions, the aqueous form is more frequently used for current washing lines. Behlau et al. [140] showed the capacity of aqueous $ClO_2$ to eliminate, immediately after treatment, *Xanthomonas citri* subsp. *citri*, a quarantine plant pathogen and the causal agent of the citrus canker. Moreover, it promoted a significant reduction of 2.4–2.8 log10 cfu/mL in the populations of live bacteria in artificially and naturally contaminated citrus fruit. Nevertheless, the industrial application of $ClO_2$ is still limited by the long time exposure (10 min to 2 h) necessary to obtain the desired microbial load reduction, the toxicity at high concentrations and its unstable nature at high temperatures, which can cause explosions [66].

Hydrogen peroxide ($H_2O_2$) is an antimicrobial agent that exerts its activity by causing oxidative stress in a broad spectrum of microorganisms. $H_2O_2$ was authorized by the U.S. Food and Drug Administration (FDA) for its use in the food industry in washing water or sterilising packaging devices and hygienic filling machines [141]. As a sanitiser of surface pathogenic microorganisms of fruit and vegetables, $H_2O_2$ can reduce microbial populations through the generation of cytotoxic agents (hydroxyl radical) before rapidly decomposing into oxygen and water with heat release [142]. Recently, $H_2O_2$ has been applied in the postharvest control of fungal pathogens in fresh fruit and to extend shelf life during storage. Nevertheless, efficiency problems have been encountered when a pure hydrogen peroxide solution was used on account of its instability [143]. As proven by Meng et al. [143], $H_2O_2$ stabilized with silver ions ($Ag^+$) provides a highly effective disinfectant activity on orange fruit pericarp and significantly reduces decay incidence after 60 days of cold storage.

Peracetic acid (PAA), also known as peroxyacetic acid, is a non-chlorinated sanitizing agent with an antimicrobial power similar to or greater than that of sodium hypochlorite. PAA poses low risks for human health and is accepted for use in organic crop production, processing and postharvest handling [144]. The inactivation capacity of PAA is based on the rapid oxidation of the cellular components of a broad spectrum of microorganisms through the production of ROS that generate instability in biomolecules, such as DNA, lipids and proteins, that are vital for the correct cellular functioning of pathogens [145,146]. PAA is usually applied in the postharvest phase of fruit and vegetable production by spraying and dipping, breaking down into acetic acid and oxygen soon after treatment [147]. PAA is used as a sanitiser in the washing process of citrus fruits [148] to reduce microbial load. Taverner et al. [149] demonstrated the efficacy of PAA in making non-viable *Penicillium digitatum* conidia after 480, 120 and 30 s exposure to warmed (35

°C) aqueous solutions of 72, 108 and 216 mg/L PAA, respectively. Lanza et al. [150,151] reported that applications of PAA at 800 μg/mL for 1 min effectively reduced green mould incidence in lemon fruit.

Ozone is triatomic oxygen and is formed by the combination of oxygen-free radicals with molecular oxygen. It is a powerful natural oxidative agent against bacteria, fungi, viruses and fungal spores [152,153]. Ozone was approved by the FDA in 2001 as a sanitising agent for direct-contact use on fresh produce and various fresh-cut fruits and vegetable, owing to its high reactivity [154]. It can be applied as a gas at a limiting concentration of 0.25 ppm [155] and as ozonated water at concentrations variable from 0.5 to 20 ppm [156]. A considerable benefit of ozone is its fast decomposition into oxygen, which leaves no chemical residue on treated products. Nevertheless, because of the lack of stability, it must be produced on-site when required. Ozone is most often generated by the passage of air or gaseous oxygen through a high-voltage electrical discharge that produces ozone molecules from oxygen atomic radicals [157]. Postharvest ozone treatments of fruit and vegetables can promote adverse effects due to their high moisture contents, enzymes and phenolic compounds. Thus, a monitored ozone concentration is needed, as well as an optimization of the treatment condition for each product [158]. Di Renzo et al. [159,160] designed a prototype to control the ozone concentration during Tarocco, Ovale and Valencia orange (*C. sinensis* (L.) Osbeck) washing using a feedback control system equipped with high-precision measuring sensors. The results revealed a low control efficacy when using ozonated water, probably due to variable degrees of contamination because of citrus fruit impurities. Strano et al. [161,162] found that aqueous ozone treatments significantly reduced total microbial count in Tarocco oranges at 3 ppm and in Clementines at a concentration of 6 ppm, in addition to extending fruit shelf life during storage at 4 °C and RH 90%. Ozone was also influential in removing off-flavours, mycotoxins and pesticide residues [163]. An effective ozone depletion system is essential to ensure workplace safety, since over-exposure to ozone (>1 ppm) may cause damage to the respiratory tract, lungs and eyes of humans [157].

Electrolyzed water (EW) is a sanitiser generated by passing a diluted saline solution, usually NaCl, through an electrolytic cell containing positively (anode) and negatively (cathode) charged platinum electrodes. By the dissociation of NaCl solution, active chlorine molecules, such as hypochlorous acid (HClO), hypochlorite ion (ClO$^-$) and chlorine gas (Cl$_2$), are produced [66]. Acidic and alkaline EW can be generated using the same production machine. Acid EW (pH 2.3–2.7), with high oxidation–reduction potential, is produced on the anode side of the electrophoresis process, while alkaline EW (pH 10–13; low oxidation–reduction potential) is produced on the cathode side. Where anode and cathode are not separated by a membrane, neutral (pH 7) or slightly acidic (pH 5.0–6.5) electrolyzed water is produced [164]. Despite its higher effectiveness, acid EW has limited application because it is corrosive, whereas chlorine gas can be released during EW production [165]. EW can be used for fresh fruit, vegetables and minimally processed products due to its strong antimicrobial effect against a wide range of microorganisms, such as viruses, bacteria, fungi and spores [165,166]. Youssef and Hussien [167] reported a significant reduction in *Penicillium digitatum* and *P. italicum* growth in in vitro tests on artificially inoculated Valencia late oranges (*Citrus sinensis* L. Osb.) and naturally infected fruit treated with EW. The salts sodium metabisulfite, potassium sorbate, potassium carbonate and sodium chloride were individually integrated (1% *w/v*) with acid and alkaline EW (water flow rate 6 L min$^{-1}$; current intensity 13A) to enhance biocidal efficacy. Naturally infected treated fruits, stored at 6 ± 1 °C (RH 90–95%) for 45 days and assessed for physicochemical parameters, confirmed the lack of detrimental effects on fruit quality.

### 3.2.2. Inorganic and Organic Compounds

Inorganic and organic compounds belong to the GRAS (Generally Regarded as Safe) category, as they are considered harmless to human health and to have minimal environmental impact, with low or undetectable residues due to their rapid degradation

immediately after treatment application [168]. GRAS inorganic or organic salts, such as sodium carbonate and bicarbonate, potassium sorbate and sodium silicate, are frequently used in citrus packinghouses, owing to their low cost, high solubility in water and the possibility of using them with other alternatives in integrated treatments [162,169]. GRAS substances are mainly known for their antifungal properties [170,171] and as resistance inducers. Youssef et al. [172] investigated the ability of sodium carbonate and bicarbonate to activate defence mechanisms in citrus fruits against postharvest green mould caused by *Penicillium digitatum*, confirming the increased activity of the enzymes β-1,3-glucanase, peroxidase and PAL in orange tissues.

### 3.2.3. Natural Antifungal Compounds

Several plant- and animal-derived substances with antimicrobial properties have recently been assessed for controlling fruit rots, with both preharvest and postharvest applications. The application of plant extracts as potential fungicides alone or in combination with other control measures is quite promising due to their well-documented antifungal activity, low phytotoxicity, decomposition and low environmental toxicity [173–176].

Several authors have reported the ability of aqueous or organic solvent extracts from different plants to control postharvest citrus decay thanks to their secondary metabolite contents, such as flavonoids, terpenes, alkaloids, ethanol, methyl salicylate, jasmonates, allicin and isothiocyanate [176–183]. Low doses of jasmonates (jasmonic acid and methyl), cinnamaldehyde [177–180], citronellal [178,181], garlic [182] and isothiocyanates [183] showed different effective antifungal activities against the major postharvest pathogens in citrus fruit, reducing the growth of *P. digitatum* (green mould), *P. italicum* (blue mould) and *Galactomyces citri-aurantii* (sour rot) in in vitro and/or in wound-inoculated fruit and/or in naturally infected fruit (Table 4).

Propolis proved effective in reducing green mould (*P. digitatum*) and blue mould (*P. italicum*) incidence in wound-inoculated fruit and naturally infected fruit [184].

Extract of *Solanum nigrum* [185] showed in vitro antifungal activity and preventive antifungal efficacy in artificially wounded lemon fruit. *Cistus* plant extracts [186] were able to inhibit the growth of fungal isolates of *G. citri-aurantii* in infected mandarins.

Pomegranate (*Punica granatum* L.) peel extract [187–189] demonstrated strong efficacy in postharvest treatments of citrus fruit due to its high antioxidant activity and antimicrobial capacity correlated with its high content of phenolic compounds [190]. In mandarins (*Citrus reticulate* var. *Kharo*) treated with pomegranate peel extract (PGE 100%) and stored at 4 °C for 20 days, decreased decay incidence (16.3%) was observed in the untreated fruits (39.6%). Moreover, the weight loss of treated fruits at the end of storage was significantly lower (14.71%) than that of the untreated control (42.28%), and the qualitative traits (firmness, pH, SST, acidity) and sensory properties of the fruits were preserved during storage [187]. The treatment of lemon fruits with 12 g/L of pomegranate peel extract (PGE) allowed the control of natural postharvest infections [189]. The application of coatings (chitosan and alginate), combined with 1% PGE, proved effective in maintaining guava fruit quality under cold storage and extending fruit shelf life for up to 20 days [190].

The Moroccan medicinal and aromatic plants *Thymus leptobotrys*, *Cistus villosus* and *Peganum harmala*, used as plant powders at 10% (w v⁻¹), inhibited the growth of *P. digitatum*, *P. italicum* and *G. citri-aurantii*. In contrast, the powder of *Eucalyptus globulus* effectively inhibited the mycelial growth of *P. digitatum* and *G. citri-aurantii* [176,191].

Decay control was also studied using essential oils, such as thymol, cinnamaldehyde, citral, eugenol, limonene, etc. [192]. Although promising antifungal activity was observed in in vitro tests, lower effectiveness was observed in relation to fruit and rind phytotoxicity [193]. Promising results have been obtained with peptides and small proteins with antimicrobial properties against various microorganisms [194]. High

purification costs, low stability and problems related to non-specific toxicity of the molecules still represent significant obstacles to their practical application [195].

**Table 4.** Efficacy of plant extracts on postharvest decay control in citrus fruit.

| Extracts | Fruit Tested | Target Pathogens | Significant Results | References |
|---|---|---|---|---|
| Cinnamaldehyde | Mandarins | *P. digitatum*, *Galactomyces citri-aurantii* | Strong antifungal properties; green mould and sour rot (*G. citri-aurantii*) reduced incidence; induced defence responses in citrus fruit | [178–181] |
| Citronellal | Oranges | *P. digitatum* | Reduced postharvest incidence of green mould in citrus fruit | [179,182] |
| Garlic | Oranges | *P. digitatum*, *P. italicum* | Higher increased activity in mixed garlic extracts with oils | [183] |
| Isothiocyanates | Mandarins | *G. citri-aurantii* | Antifungal properties both in vitro and in vivo conditions | [177,184] |
| Propolis | Mandarins | *Penicillium digitatum*, *P. italicum.* | Reduced green mould (*P. digitatum*) and blue mould (*P. italicum*) incidence in wound-inoculated fruit and naturally infected fruit | [185] |
| *Solanum nigrum* | Lemons | *P. digitatum* | In vitro antifungal activity Preventive antifungal efficacy in artificially wounded fruit | [186] |
| *Cistus* plant extracts | Mandarins | *G. citri-aurantii* | Antifungal properties in both in vitro and in vivo conditions | [187] |
| Pomegranate (*Punica granatum* L.) peel extract | Lemons | Primary postharvest pathogens in citrus fruit (in vitro tests) | Strong efficacy of in vitro and in vivo treatments due to the high content of phenolic compounds | [188–191] |
| *T. leptobotrys*, *C. villosus*, *E. globulus* and *P. harmala* extracts | - | *P. digitatum*, *P. italicum*, *G. citri-aurantii* | High antifungal activity in in vitro tests | [177,192] |

### 3.2.4. 1-MCP

In citrus fruits, many physiological disorders (CI, SERB, pitting) have been associated with ethylene production [196], which has been shown to dramatically accelerate chlorophyll degradation, fruit softening, volatile aroma biosynthesis, abscission and browning of attached leaves [197,198]. The ethylene perception inhibitor 1-Methylcyclopropene (1-MCP) interacts with ethylene receptors and prevents ethylene-dependent responses.

Combined with other materials for handling and then mixed with a specific amount of water or other solution to release it into the air, 1-MCP is used in closed off-sites, such as coolers, truck trailers, greenhouses, storage facilities and containers, to maintain the freshness of ornamental plants and flowers and reduce the ripening rate of climacteric fruits [199].

However, controversial information has been collected on the physiological and quality responses to 1-MCP of harvested citrus fruit, suggesting that the effect of concentration and cultivars on the efficacy of 1-MCP in citrus fruit cannot be neglected.

For instance, McCollum and Maul [200] have reported that 1-MCP inhibited de-greening but stimulated respiration and ethylene biosynthesis in grapefruit. Similarly, 1-MCP treatment blocked the de-greening process but increased chilling injury symptoms,

decay development and the accumulation of volatile off-flavours in 'Shamouti' oranges. It also significantly inhibited leaf abscission and chlorophyll degradation in Shatangju mandarins (*Citrus reticulate* Blanco) [198]. Several authors [201] have reported that the application of 1-MCP reduced CI and peel pitting in 'Fallglo' tangerines and grapefruits when applied at concentrations between 50–500 µg L$^{-1}$, while concentrations higher than 1 mg l$^{-1}$ enhanced the development of decay in 'Fallglo' tangerines and white 'Marsh' grapefruits.

In oranges, the application of 1-MCP markedly reduced the incidence of SERB and increased the development of decay when the 1-MCP concentration used was high (1–5 mg L$^{-1}$). Lower concentrations (0.5 µg L$^{-1}$) were adequate to reduce CI incidence in 'Nova' and 'Ortanique' mandarins [202]. Overall, 1-MCP should be used on a product-by-product basis, with treatment temperature, concentration and lastingness of effect being determined in relation to efficacy.

### 3.3. Biocontrol

Biological control by microbial antagonists (bacteria, yeasts, fungi) is still under investigation as an alternative to the application of synthetic fungicides for disease control in the field and postharvest applications [8,203–206]. Wang et al. [8] provided an overview of representative recent research on antagonists used to manage postharvest fungal decay in citrus fruit. In a very recent study, Zhang et al. [205] reported the antifungal activity results for the fermentation product of the endophytic fungus *Aspergillus aculeatus* GC-09 (minimum inhibitory concentration (MIC) = 0.3125 mg/mL) used to combat the spore germination and mycelial growth of *P. italicum.*

Yeasts remain the most preferred antagonists for postharvest treatments of fresh fruit, applied by dipping or sprayed directly onto fruit or used as ingredients in edible coatings, and are usually selected from natural sources [204,207,208]. The efficacy of yeasts results from a mechanism of competition against fungal pathogens for nutrients and space at wound sites. In addition, they act through other strategies, such as mycoparasitism, induction of resistance in the host tissue and production of volatile secondary metabolites [8,209,210]. As recently remarked by Zhu et al. [211], the antagonist yeast *Yarrowia lipolytica* displayed a great ability to adapt to the mandarin wound environment at storage temperatures of 20 and 4 °C, better than *P. digitatum* and *P. italicum.*

Commercial products, such as Aspire™ (*Candida oleophila*), Pantovital™ (*Pantoea agglomerans*) and Biosave™ (*Pseudomonas syringae* Van Hall), were first registered in the United States and Spain for the postharvest control of *Penicillium expansum* in apples and green and blue mould in citrus fruit. Aspire is not commercialised, and Biosave was subsequently extended to cherries and potatoes. Shemer™ (*Metschnikowia fructicola*), initially registered in Israel for both pre- and postharvest application in various fruits and vegetables, was later acquired by Bayer CropScience (Leverkusen, Germany) and recently sublicensed to Koppert (Uithoorn, Netherlands).

Although most microbial antagonists have shown promising results in reducing the incidence of fungal rots in laboratory-scale experiments, their commercial application in citrus packinghouses is still limited because of their lack of curative activity, high cost, and lower level of control efficacy when applied as stand-alone treatments [151,212]. Therefore, integrated strategies to enhance biocontrol performance are needed [213,214].

Recent studies concerning the development of new biocontrol agents have suggested that microbial communities (microbiomes) in and on plant tissues should be further investigated for the development of formulations of microbial consortia, rather than single antagonists, to be used in preventing postharvest diseases and physiological disorders [8,215].

### 3.4. Coatings

A commodity's visual appearance is a powerful feature in determining market acceptance as it is the first attribute buyers notice. However, it is not always related to

internal quality and taste. Consumers are attracted to fresh produce with a shiny appearance and are more likely to buy it. Over the years, various types of waxes have been used to improve the cosmetic characteristics (gloss and colour) of fruit and vegetables and decrease fruit weight loss and chilling injuries by reducing transpiration and respiration. Commercial waxes commonly used in citrus packinghouses are anionic microemulsions that contain resins and waxes, such as shellac, carnauba, beeswax, polyethene and petroleum, often applied in combination with synthetic fungicides [5]. In recent years, alternative natural-origin coatings have been investigated for postharvest applications to replace synthetic waxes [216–218]. These coatings can be applied on the surface of fruit (edible coatings) or placed on packaging (edible films) and can also be supplemented with antimicrobial compounds [219,220]. Both provide a semi-permeable barrier to water vapour, oxygen and carbon dioxide between the fruit and the surrounding atmosphere, thus preventing chemical and microbiological spoilage and extending the product's shelf life [221].

Several novel edible coatings applied alone or as carriers of antimicrobial compounds and/or biological agents (antifungal ECs) have been evaluated for citrus fruit, with promising results for postharvest disease control and fruit quality preservation [222–227]. Saberi et al. [224] demonstrated that composite edible coatings based on pea starch and guar gum blended with a lipid mixture proved effective in reducing the respiration rate, ethylene production, weight and firmness loss and decay rate of coated Valencia oranges stored for up to 4 weeks at 20 °C and at 5 °C. Ali et al. [226] showed that the application of edible coatings containing 1% carboxymethyl cellulose (CMC) on 'Kinnow' mandarins stored at 5 ± 1 °C for 30 days reduced both chilling injury symptoms (1.68-fold lower in coated fruit with respect to the control) and disease incidence (2.15-fold lower in coated fruit with respect to the control) at the end of storage. Strano et al. [227] reported the effectiveness of a novel pectin-based edible coating combined with the antagonistic yeast *Wickerhamomyces anomalus* BS91 in reducing *Penicillium digitatum* decay incidence by up to 90% in artificially inoculated Tarocco orange fruit.

Salem et al. [228] displayed results obtained for the application in vitro and on Navel oranges infected with *P. digitatum* of a new nanocomposite (NCT/PPE/SeNPs) comprising biosynthesized selenium nanoparticles (SeNPS) with pomegranate peel extract (PPE) and chitosan nanoparticles (NCT). The results of the in vitro tests carried out with the direct application of NCT/PPE/SeNPs on *P. digitatum* mycelia showed a remarkable lysis and deformation of fungus hyphae within 12 h of treatment, whereas a 100% control of green mold infection was obtained in coated fruit.

## 4. Non-Destructive Methods for Quality Assessment

In recent years, the citrus industry has increased the need for fast, accurate and non-destructive tools for online/inline fruit quality assessment, monitoring and early detection of pathological decay. The research focused on developing and applying novel non-destructive methods to assess fruit quality with rapid, accurate measurements of fruit without causing fruit or chemical waste [229].

In modern citrus packinghouses, advanced systems for sorting and measuring the external and internal quality of large amounts of citrus fruits rely on automated and reliable inspection systems based on computer vision techniques, equipped with electronic sorting devices capable of examining fruit images at a very high speed and measuring external properties, such as colour, size and the presence of damage or defects [230].

### 4.1. Visible and Near-Infrared Reflectance Spectroscopy (Vis/NIR)

Aside from the investment costs for the equipment, Vis/NIR analyses do not require trained staff, chemicals, or additional materials, allowing the definition of these techniques as sustainable or "non-polluting". The ease of management and the high accuracy of the tools composing the Vis/NIR equipment (e.g., optical fibers) make this technique suitable for constructing at-line, online and inline sensors in production plants

to control industrial processes and food quality. However, the accurate prediction of an unknown measure depends on the calibration model used. If an algorithm does not describe with very high accuracy the correlation occurring between the wavelengths and the independent variable, it cannot be exploited on a further batch. For example, Matera et al. [231] tested the accuracy of Vis/NIR techniques in detecting the Imazalil content in water solution. Imazalil is a fungicide used to control postharvest losses due to *Penicillium* spp. in citrus by dipping the fruits in an aqueous solution containing Imazalil. Its concentration, related to the amount of fruit treated throughout the day, can be dramatically reduced. Therefore, the efficacy of a spectroscopy-based control system was evaluated to monitor fungicide concentration online during treatment. The results showed that not only was a different accuracy level obtained when using Vis rather than NIR (highest accuracy with NIR) but also that data pretreatment (such as normalization, first or second derivative degree) hugely affected the prediction capabilities of the algorithms (PCR, PLS, SVM, ENSEMBLE).

Vis/NIR techniques applied to fruits have some drawbacks [232]. When a light beam hits a fruit or any other biological sample, the incident radiation may be specularly reflected, absorbed or transmitted. The relative contribution of each phenomenon depends on the item's light-scattering properties related to the sample's microstructure and chemical composition [233].

In thin-skinned fruits, most of the light beam interactions occur on the flesh, and the skin has mainly a modulation effect upon the spectra. In contrast, in citrus fruits, the interactions occur between flavedo and albedo, and few photons probe the flesh. Considering this, amongst the Vis/NIR spectra-acquiring methods (transmittance and reflectance), in citrus, the reflectance mode is easier to handle, with higher signal levels than the transmittance mode [234]. However, the assessment of quality indexes in citrus fruits through Vis/NIR spectroscopy depends on the interplay between pulp and skin biochemistry and their optical properties, and, to date, the efforts to correlate optical properties and quality indexes have been evident only for titratable acidity, total solid soluble content, pH and firmness [235–237].

### 4.2. Hyperspectral Imaging Analysis

Hyperspectral imaging analysis (HSI) is a non-destructive optical analysis technique. However, unlike other optical technologies that can only scan for a single colour, HSI can distinguish the entire colour spectrum in each pixel, simultaneously considering the spectral and spatial information of samples. In HSI, each image pixel contains spectral information, which is added as a third dimension of values to the two-dimensional spatial image, generating a three-dimensional data cube, sometimes referred to as a data hypercube or an image cube [238]. Amongst spectral imaging, hyperspectral imaging has been widely used to estimate food quality in terms of surface damage (bruises, chilling and insect bites), surface quality (firmness, moisture content, hardness, rottenness and fungal presence) and assessment of some biochemical components in vegetable products [239].

HIS has also been exploited for citrus fruits analysis, with successful results in the detection of external defects in oranges, such as insect damage, wind scarring, thrips scarring, scale infestation, canker spot, copper burn, heterochromatic stripe, phytotoxicity and stem-end rot [240]; citrus black spot in 'Valencias' oranges [241]; pectin content in 'Lanelate' oranges [242]; and solid soluble content [243] and pesticide residue [244] in 'Navel' oranges. As for Vis/NIR analysis, data pretreatment prior to calibration is also required for HIS to leverage the data set.

The direct model calibration building suffers from some drawbacks, as some peaks or wavelengths in the spectra are convoluted due to both the chemical complexities of foodstuffs and errors in the measurements (light scattering, temperature drift, electrical noise).

Some preprocessing techniques are used to remove irrelevant information that can degrade the performance of the numerical algorithm used to develop the calibration model.

These include MSC (Multiplicative Scattering Correction), SNV (Standard Normal Variate), first and second derivatives filters (Savitzky–Golay), de-trending, scaling, and normalization [230]. Several attempts using HIS have been successfully carried out to discriminate between 11 types of pathological decay in citrus, corresponding to a classification success rate of around 89% for detecting rottenness [245]. The findings carried out by the investigation mentioned above suggest that imaging technology coupled with multi-sensors (temperature, humidity, $CO_2/O_2$) [246] can be an effective tool for the effective control of safety and quality across the range of citrus fruits.

### 4.3. Raman Spectroscopy

Raman spectroscopy is a non-invasive optical technique that is easy to conduct and only requires a very compact set-up, which makes it portable. Raman spectroscopy measures inelastic light scattering based on a monochromatic source, providing information on the chemical composition by recording the molecular vibrations of the constituent components (spectral footprint). It can detect subtle molecular and biochemical changes in tissues [247] and can be conducted on living organisms, allowing the characterization of the chemical structure of tissues and distinguishment between normal and diseased tissues [248–250]. Liu et al. [251], using this technique coupled with partial least squares discrimination analysis (PLS-DA), reached a correct recognition rate of 100% in detecting citrus greening bacterial disease (HLB), the most severe citrus disease caused by phloem-limiting bacteria.

### 4.4. Nuclear Magnetic Resonance

One of the most critical features of NMR is measuring water content and distribution, which is very useful for assessing ripeness, defects or decay in fruits and vegetables. High-resolution images can contribute to the evaluation of maturity and quality parameters. However, they can also help the understanding of physiological processes, such as water transport and the presence of water-soluble metabolites [252].

NMR can be applied to a wide range of liquid and solid matrices without altering a sample or producing hazardous wastes.

Chen et al. [253] showed the reliability of NMR imaging in detecting textural changes under various conditions, such as dry regions in 'Washington' oranges and 'Valencia' oranges; bruising in apples, peaches, Asian pears and onions; worm damage and stage of maturity in Asian pears, red/green tomatoes and pineapples; the presence of void spaces in potatoes, cucumbers and tomatoes; and the presence of seeds and pits in olives and prunes. Butz et al. [254] also reviewed the use of NMR to measure water states and water-related properties of fruits, including apples, pears, peaches, kiwi fruits and oranges. Therefore, there is ample evidence of the applications of NMR for assessing or inspecting quality parameters of a variety of fruits, such as ripeness, defects and decay, as well as differentiating between unaffected tissue, brown tissue and cavities during various conditions, such as postharvest, storage and transportation.

In addition to its ability to assess the physical properties of certain foods, NMR can also be used to investigate chemical attributes related to volatile organic compounds (VOCs) [255].

NMR spectroscopy combined with statistical analysis techniques is one of the promising tools for food quality control. One such combination is Spin Generated Fingerprint Profiling (SGFP) with NMR, which fully automatically performs sample transfer, measurement, data analysis and reporting for quality control of fruit juices [256]. Having established a spectral database, NMR enabled the simultaneous identification and qualification of 28 different compounds in a mixture. It also allowed the detection of fraudulent addition of sugar, citric acid, lemon juice and galacturonic acid (an indicator

of an exhaustive enzymatic treatment). SGFP represents a heterogeneous collection of statistical models which can be applied consecutively to one single spectrum, such as specific models for multi-fruit type separation, fruit type differentiation between citrus varieties (e.g., citrus sinensis and citrus reticulata), differentiation of product categories (e.g., orange juice and orange juice made from concentrate) or the characterization of compositional differences between two groups of similar products [257].

### 4.5. Nanosensors for Early Detection

By deploying the above-mentioned spectroscopy- and image-based analytical techniques (NIR/VIS, HIS, Raman and NMR) coupled with proper image processing techniques, it has been possible to correlate optical properties with several parameters to detect or predict the behaviour of many quality parameters, mainly chemical and physicochemical, and, in some works, the extent, at an early stage, of pathological decay in oranges [258–262] or freezing damage in sweet lemons [263] and oranges [264,265].

Early detection of a pathological condition has many advantages, such as early warning for planning actions to mitigate colonization and thus postharvest waste and loss of crops.

In the last few decades, many advances have been made in relation to quickly and accurately detecting hidden fungal colonization in *Citrus*, in the field, during handling or storage early on. These advances encompass the development of nanosensors for the identification of fungal attack before the appearance of visible signs of infection on citrus surfaces using an array composed of electrochemical or biological receptors.

Various nanosensors are being developed to meet the different requirements in food quality inspection and food processing control. The most common nanosensors for food and agriculture applications include optical sensors, electrochemical nanosensors, e-noses, e-tongues and biosensors [266].

For instance, a range of VOCs accompany upcoming pathological decay in citrus fruit, such as limonene, b-myrcene, a-pinene, sabinene, acetaldehyde, ethanol, ethylene, and $CO_2$, which can be detected to follow the extent of the decay [9].

VOCs were easily distinguished using an electrical sensor array for early-stage on-farm detection of HLB [267].

Limonin is a VOC responsible for the bitter taste of citrus fruits, such as oranges, grapefruits, etc. An abnormally high level of limonin indicates HLB, which results in stunted tree growth and affects fruit quality in terms of nutritional value, taste, texture and aroma. Ceria nanoparticles are microsensors based on the quantification of limonin to detect HLB using an organic electrochemical transistor platform [268]. As it is challenging to identify HLB-infected trees because they may remain asymptomatic for months to years after infection, the finding of this study is particularly interesting as the sensor can be exploited to correlate VOCs with the decay index at harvest, which is relevant to deciding postharvest destinations and treatments of fruits.

The electronic nose (e-nose) relies on a portable device equipped mainly with broad-spectrum chemical sensors. With the support of a pump, the sample's headspace is passed over the e-nose's detector, which generates signals acquired by a computer and processed by pattern-recognition algorithms, providing a fingerprint of the volatiles present in the analyzed sample [269].

Research on citrus fruits using e-nose technology has been concerned with the identification and characterization of different citrus cultivars and varieties, postharvest quality monitoring, disease identification through the detection of specific volatile biomarkers in the early stages of deterioration and metabolic changes as a result of fruit respiration, transpiration and/or fermentation during storage [270–272].

The aromatic variation of oranges was successfully studied for one month using e-nose, principal component analysis (PCA) and partial least squares discriminant data analysis (PLS-DA). The study produced evidence of the excellent sensitivity and

resolution of e-nose sensors in measuring aromas of decay in oranges and predicting storage days correctly [273].

However, although postharvest fungal disease detection in citrus fruits under cold storage conditions is crucial, few works have been carried out at present. For instance, the e-nose may be successfully applied as a reliable, non-destructive technology to identify *P. digitatum* in 'Valencia' oranges during cold storage [270,274] and to discriminate between uncontaminated lemons and lemons contaminated with *P. digitatum* spores [275].

The electronic tongue (e-tongue) is a multisensory device for the rapid qualitative assessment of different liquid food products based on chemical sensor arrays, a signal trapping device and software which converts the signals into appropriate results. Among its various uses, e-tongue has excellent potential for studying the shelf life of different products. A study carried out on Satsuma mandarins [276] showed that the e-nose and e-tongue systems, combined with some algorithms, could provide a fast and objective detection system to trace fruit quality and, in particular, to discriminate different ripening stages and trace internal quality changes (i.e., ascorbic acid, soluble solids content, total acid and sugar/acid ratio). Raithore et al. [277] successfully used an e-tongue to differentiate between orange juice made from healthy fruit and fruit affected by HLB disease. Orange juice made from the fruit over the harvest season and from fruit harvested from healthy or HLB-affected trees were separated by harvest maturity, disease state and disease severity using an e-tongue system.

Biosensors rely on a biologically active element, such as an antibody, enzyme, oligonucleotide or receptor blocked on a specific array, which is highly selective due to the possibility of tailoring the specific interaction with the analyte. The interaction analyte bioelement produces a biosignal (optical, electrochemical or colorimetric) that is converted into a measurable signal by a transducer. With respect to biosensors, extremely interesting is the capability to detect early on fungal mycotoxin presence, which is potentially harmful to public health [278]. However, little evidence of biosensors' exploitation for the early detection of pathological or physiological alterations in citrus fruits has been reported in the literature, owing to different effectiveness and sensitivity results.

Electrical bioimpedance spectroscopy measurements of the flavedo, albedo and pulp in 'Star Ruby' grapefruit were carried out for early freeze-damage detection in grapefruit, showing a Correct Correlation Rate of 100% [279]. Chalupowicz et al. [280] used a bacterial cells-based biosensor to follow up *P. digitatum* colonization in Valencia. That approach was based on bacterial luminescent responses to changes in volatile organic compounds (VOCs) following infection, enabling fungal infection detection on the third day of infection, before the appearance of visible signs.

Wang et al. [267] reported the application of electrical biosensor arrays based on single-walled carbon nanotubes (SWNTs) decorated with single-stranded DNA (ssDNA) for the detection of four VOCs—ethylhexanol, linalool, tetradecene and phenylacetaldehyde—that serve as secondary biomarkers for the detection of infected citrus trees during the asymptomatic stage of HLB.

Biosensors are particularly useful in the citrus industry and are expected to address the severe and urgent needs encountered there by providing fast, simple and cost-effective methods for detecting pathological diseases, especially those caused by unculturable microorganisms.

## 5. Future Directions

The future of citrus fruits postharvesting will be characterized by deploying new technologies to optimize treatments and follow up on fungal diseases' occurrence to control quality and reduce food losses. Alongside the need to spread pre-cooling techniques, which are still not very widespread for citrus fruit and poorly applied in many cases, the use of innovative plastic films to contain water loss during refrigerated storage will find greater application, together with innovative washing and packaging techniques,

including the use of protective atmospheres and passive modulation systems for the atmospheres inside packages. Green techniques that do not leave residues on products will undoubtedly be preferred in the washing phases. Finally, new biocontrol techniques will be able to preserve citrus fruits in the field or during handling, shipment or storage. The development of combined approaches, such as multi-sensors, advanced imaging technology analyses and IoT tools, to monitor quality markers in real time will undoubtedly improve the competitiveness of the entire supply chain.

## 6. Patents

Reference n.13 is related to the following patent: 2018—Device for controlling gaseous exchanges between the inside and outside of a container for solid or liquid food products. PCT/IB2016/0506600 (https://patentscope.wipo.int/search/en/, accessed on 24 June 2022). EU patent no.: 3303174 (26 December 2018).

**Author Contributions:** Conceptualization, M.C.S.; writing—original draft preparation, M.C.S., M.A., F.G. and A.M.; writing—review and editing, M.C.S., F.G., G.P., G.A. and G.C.D.R. All authors have read and agreed to the published version of the manuscript.

**Funding:** This paper is part of the project Sharing Knowledge to Increase Post-harvest Efficiency—"SKIPE", which is funded with the financial assistance of the European Union in the framework of the Operational Programme ERDF Basilicata 2014–2020. The document's content is the sole responsibility of UNIBAS and can under no circumstances be regarded as reflecting the position of the European Union and/or the Operational Programme ERDF Basilicata 2014–2020 authorities.

**Institutional Review Board Statement:** Not applicable.

**Informed Consent Statement:** Not applicable.

**Data Availability Statement:** Not applicable.

**Conflicts of Interest:** The authors declare no conflict of interest.

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
