# Peer review of "Postharvest Technologies of Fresh Citrus Fruit: Advances and Recent Developments for the Loss Reduction during Handling and Storage"

_horticulturae, doi:10.3390/horticulturae8070612_

Round 1

Reviewer 1 Report

The manuscript horticulturae-1711884 reviewed the advances and recent developments of research on technologies used to reduce losses in citrus fruit after postharvest and future perspectives. The following comments in details for the authors to consider.

  1. Line 13-14, references for “30-50%” is required.
  2. The title is “Postharvest technologies of citrus fruit: advances and recent developments for the loss reduction during the whole chain”. But this review mainly focuses on post-harvest preservation technologies, and does not address the entire supply chain including primary processing at origin, storage, transportation, and retail.
  3. This review focuses on postharvest issues, new alternative strategies, advanced postharvest technologies and future directions of citrus fruit, but only new alternative strategies and advanced postharvest technologies are mentioned in the last paragraph of the introduction section.
  4. Line 86-88, the order of the sentence should be adjusted. The definition of CI should be described first, and then the sensitivity of different citrus varieties and cultivars to CI should be described.
  5. Section 2.2 is inappropriately segmented. Lines 84-86 should belong to the first paragraph and lines 89-100 should belong to the second paragraph introducing chilling injury.
  6. Some paragraphs in section 4.1.1. do not have first line indents.
  7. It is not clear whether the "their" in line 476 indicates the ninth reference or some other reference.
  8. There is no mention of Table 2 in the manuscript.
  9. The first and second paragraphs of section 4.1.3 should be switched.
  10. The last paragraph of section 4.2.2.3. should add references to the application of electronic tongue on citrus fruits.

Author Response

Dear Reviewer,
thank you very much for your suggestions. We accepted all your comments and improved our manuscript. 
Best regards
FG

Reviewer 2 Report

I congratulate the authors on the multi-threaded approach to the discussed topic, from well-known issues to the latest research results in the field of quality and storability of non-climacteric fruits. 

Author Response

(The authors gave the same response as above.)

Reviewer 3 Report

Comments entitled

Manuscript ID: horticulturae-1711884 entitled ‘Postharvest technologies of citrus fruit: advances and recent developments for the loss reduction during the whole chain - review’. The study contains a lot of information about postharvest techniques can be used in the fresh fruit industry. However, the information presented in the whole manuscript is general and it lacked details information about citrus.

Please find the specific comments below.

Introduction section lack a lot of information, I would suggest the authors should include convince the review why this review is important, what are the available review literature in similar subject for citrus, which information have not been covered on those literatures (gas analysis).

Check this article and there are many more available on search engine.

Zacarias, L., Cronje, P. J., & Palou, L. (2020). Postharvest technology of citrus fruits. In The Genus Citrus (pp. 421-446). Woodhead Publishing.

Chen, J., Shen, Y., Chen, C., & Wan, C. (2019). Inhibition of key citrus postharvest fungal strains by plant extracts in vitro and in vivo: A review. Plants, 8(2), 26.

Wang, Z., Sui, Y., Li, J., Tian, X., & Wang, Q. (2022). Biological control of postharvest fungal decays in citrus: a review. Critical Reviews in Food Science and Nutrition, 62(4), 861-870.

Line 39 and 49, …’diseases’.. the right word could be pathological disorder

Line 84-85. Please replace the sentence by the following

‘suden metabolic changes such as high respiration and transpiration rate could be the result of senescence….’

Consistency of words such as disease, decay, pathogen should be followed. Introduction (section 1 and 2) had not have any idea about decay but it newly introduced at section 3. With out the consistency of the above words I mentioned and more one would not get the main idea of the review

Line 129-130. Should be supported by a reference please.

Line 179-180. I don’t see any consistency or right flow of information about this sentence.

Line 186-196. There is no application of UV for citrus fruit postharvest treatment and I don’t think its relevant to include it here unless the author added information about its application.

Section 3.1.5, section 3.1.6 there is no application of PL or ultrasound on citrus at these sections. Thought its irrelevant

In general for section 3.1, I would recommend the author to summarize all the subsections which have application on citrus fruit in a table since the information related to there application on citrus fruit is limited as presented.

I found the way the authors presented  section 3.2 is the same as section 3.1. The authors spend too much time to explain the technologies rather their use on citrus fruit is limited, mostly one study in each section. I would think there are quite lots of published reviews detailed about these technologies and the focus here should have been on their application, which this review lack that.

Section 4. Advanced postharvest technologies, this section is not convincing. One would expect to see advanced techniques in this section. However, the authors went back to the common precooling and packaging and there is no advances at any case. Furthermore, the way  the authors wrote this section as well is very general and failed to narrow it to application on citrus, how studies use this technologies and in which conditions addressed thee postharvest problem the citrus fruit industry is facing.

Section 4.2 presented as a postharvest treatment techniques rather than a postharvest quality analysis techniques, which is miss information.  In addition the use of these analytical techniques used for citrus fruit quality analysis were not addressed. As I said above how the authors presented  the postharvest technologies on citrus fruit the analytical tools are also presented in general, that there are plenty of reviews available about this tools as well. The authors should work more to show the reader how using these tools helps the postharvest studied that conventional methods, still missing.

This review contains a huge amount of information about postharvest treatment techniques and postharvest analytical techniques however, the authors missed to address the aim of the review. The information presented about citrus is very limited. Furthermore, the whole section needs rearrangements and proper explanation of sections.

Author Response

(The authors gave the same response as above.)

Reviewer 4 Report

The manuscript surveys the postharvest technologies of citrus fruit, critically reviewing "the current knowledge about the safer sustainable strategies, as well as advanced postharvest handling and storage technologies". It somewhat resembles the previous review of the same team of 2017 (ref. [4]) but includes enough new information to be acceptable as a new review.

However, significant improvement is needed before the material becomes publishable. The recommended changes are presented below, generally in the order of appearance in the text.

L.32: kumquat is now affiliated with Citrus genus (C. japonica), see Wu et al. (2018), Nature 554 (7692), 311-316.

L.33-35: as far as I understand, Tarocco orange is now spread much wider than in one location indicated. Maybe "originally grown"?

L.58: in the orchard, not "in the field".

L.61-64: sour rot, stem-end rot and gray mold are predominantly postharvest diseases, even if sometimes can develop on the fruit still attached to the tree. Presenting them exclusively as pre-harvest pathogens is misleading. Please recheck.

L.67-69: Rhizopus rot and Whisker rot (present genus name completely, not P., otherwise it can be attributed to Phomopsis mentioned above) typically do not belong to quiescent infections, in contrast to anthracnose and (not mentioned in this connotation) stem-end rot. Please recheck.

L.95-97: please keep in mind that the classical view (see Lyons, 1973, Ann. Rev. Plant Physiol. 24(1), pp.445-466) is that membrane disorganization is a primary event in the CI development, while ROS generation is one of the secondary ones. See Liang et al., 2020, Food Quality and Safety, 4(1), pp.9-14. However, the authors certainly can express a different opinion.

L.105: maybe opposite, oil gland collapse results in pitting? Pitting is a phenotypic symptom caused by oil gland collapse.

L.106-109: "Unlike CI, it [postharvest pitting] spreads…" The authors stress a non-CI pitting. It should be specified before (L. 84-94, the description of CI external manifestation) that typically pitting is a CI symptom in citrus and many other crops, but in some cases may occur at non-chilling temperatures due to other factors. See for example, Petracek et al., 1995. Pitting of grapefruit that resembles chilling injury. HortScience, 30(7), pp.1422-1426.

Section 3.1.1. 'Heat treatments': The section is incomplete and ignores an extensive information being a part of "current knowledge" mentioned in the review objectives.  Discussing hot air treatments of citrus, the authors may want addressing the earlier works, e.g. Ben-Yehoshua et al., 1987. Postharvest curing at high temperatures reduces decay of individually sealed lemons, pomelos, and other citrus fruit. Journal of the American Society for Horticultural Science 112(4) 658-663. Hot water treatments are completely ignored in the review, in spite of their extensive research and commercial implementation. See, for example, Schirra, M., D'hallewin, G., Ben-Yehoshua, S. and Fallik, E., 2000. Host–pathogen interactions modulated by heat treatment. Postharvest Biology and Technology, 21(1), 71-85, and Palou, L., 2013. Mini-review: heat treatments for the control of citrus postharvest green mold caused by Penicillium digitatum. In: Microbial pathogens and strategies for combating them: science, technology and education, (A. Méndez-Vilas, Ed.) pp.508-14.

Sections 3.1.2 and forth: On the contrary, other sections are, to my opinion, extended disproportionally to their real proven contribution to the area. For example, out of all references in the section 3.1.2 Photosensitisation, none address citrus fruit and only one is relevant to citrus industry, but pre-harvest (citrus canker). In many cases, the authors mention by themselves the limited efficacy of approaches and still devote a lot of manuscript space to their description.

Presenting the light treatments (UV, LED, pulse etc.), keep in mind (and mention) their low penetration capacity and therefore the need to solve shading problems in order to implement these approaches. It may be especially problematic if durable exposure (e.g. 2 days in ref. [44]) is required.

L.321: ref. [93] and not [9].

Section 3.2.2. 'Free residues compounds': what does it mean? Maybe "residue-free"? It may be more acceptable grammatically, but still I am not sure that it is correct factually. For example, ozone leaves virtually no residues, but it is presented in a different section. The section deals with inorganic salts. Are you sure they do not leave residues? Maybe, the section can be called simply "Inorganic compounds"?

Section 4.1.1. Precooling: it may be reasonable to mention, as in the previous review of the authors [4] that room cooling is still the most popular method with citrus fruits, mainly due to the economic reasons.

Section 4.1.2. Controlled and modified atmosphere storage: this section is controversial and poorly organized. The benefits of the MA and CA for citrus fruits are claimed (e.g. L. 489-493) and then denied (L. 497-499; 511-514), but then presented again in Table 2 as beneficial atmosphere compositions. BTW, no reference to this table was found in the text.

The format of this review does not allow educating the reader with general knowledge; assume that the review is addressed to a prepared reader and limit yourself with examples of citrus-related studies (this recommendation is true also with other sections). I would recommend first presenting the physiological responses of various citrus fruits to oxygen and carbon dioxide, and further illustrate these effects by examples from specific studies with citrus fruits kept under MA and CA conditions. I would recommend presenting an approach of superatmospheric oxygen storage in citrus (e.g., Molinu et al., 2016).

It is strange that such broad review ignores the topic of 1-MCP applications to citrus.

L.562: I would recommend separating the subsection 4.2 into a special section 5 because it has a different subject. It addresses fruit quality evaluation methods while until this point, the review surveys fruit preservation techniques.

This section is disproportionally broad. I would recommend adhering to specific examples of applying those or other techniques to citrus fruits.

The review needs language edition; it contains many problematic phrases and expressions. Some examples can be found in lines: 12, 36, 38, 40, 78-79, 184-185, 275-276, 487-489, 557, 563-565, 765.

Author Response

(The authors gave the same response as above.)

Round 2

Reviewer 3 Report

Comments entitled

Manuscript ID: horticulturae-1711884 - Revised Review

Title: Postharvest technologies of citrus fruit: advances and recent
developments for the loss reduction during the whole chain

General comments

·       The most important issue the authors should include here is what was their methodology, where did the collection of the articles they used comes from, which platform did they use (MDPI, Science Direct, Scopes, Nature, or its just every article from everywhere on the internet). 

·       The authors tried to cover a vast area of research interest and in between they failed to explain or write even a single section properly. They missed important information they have to include rather they focuses on general statement. These affects the quality of the manuscript. The authors didn’t make an argument why they need to write this review, they didn’t specify how this review is different from the other existing reviews which were written in detail. 

·       In general, the presentation of most of the paragraphs should be corrected. It is important to avoid using one sentence as a paragraph and it is important to make a connection between different paragraphs. 

·       The authors should work and reduce the plagiarism percentage of this manuscript. There are many direct copies and paste without a reference than using their own words

Specific comments

Table one, Table three - reference/s

Section Two, ‘title;  Postharvest issues’ could be better if it changed to a proper one may be ‘Causes of postharvest loss in citrus fruit’

Section 3.1.1. is there a study done on citrus fruit using heat treatment?

Line 185.  Which storage temperature did they use and how long was the storage duration?

Line 232. How much is extremely high dose rate? And few milliseconds means?, and

Line 235. How much is less time treatment means ?

Line 243. Please include the fruit type

Line 341-343. How long did the storability of the fruit improved, what type of cold plasma system did they use, what was the duration of the treatment, how much was the out put discharge, cultivar of mandarins?

Line 346. please add the output discharges and treatment time  

Line 348. Was the study done by Pankaj for the same fruit type with Aakudo and Yagyu, if not how does the argument made here. Does lipid oxidation applicable for fruit. I would think the authors focus on the effect of CP on fruit quality parameters rather.

Section 3.1.4 and 3.1.5 since there is no study on this technologies for citrus fruit, the authors can only present a summery of this technologies application on a table or in a statement in the appropriate section rather than making a sub section for them.

Line 552-553. Reference?

Line 554-556. Which citrus fruit, are the two studies did on the same fruit type and cultivar? How much was the concentration of PAA, duration of application PAA, how many log reduction is this?

Line 57,. Cultivar please?

Line 587. …basic EW.. please change this to alkaline (more appropriate)

Line 587-588. The statement ‘Acidic and basic EW can be generated using different production machines’ is not correct. They both can be produce in one machine but collected from the Anode and Cathode side of the electrophoreses process. Please correct it.

596-599. What type EW was used in this article please, how much was the ratio of EW and salt solution, how many of Penicillium species did they use in vivo or in vitro?. Please include information

Line 636-642, what is the relevant of this section. There are many different natural antifungal compounds but would be relevant to incorporate them if they have been used in citrus fruit or otherwise

Line 643-645. Very good information but what would the readers get from this since it’s a very general sentence rather the authors could give detail information such as as follows,

The Moroccan medicinal and aromatic plants T. leptobotrys, C. villosus, E. globulus and harmala, used as plant powder, resulted in a higher antifungal activity against the main postharvest pathogens of citrus’

 It would be very informative if the manuscript contains information related to the concentration of the powder, the application method, the application duration, the type, and cultivar of the citrus plant, include also if there was a storage condition.

This supposed to be how the information from different articles supposed to be presented throughout the manuscript but the authors uses a very general sentence rather.

In line 672-674, The authors them sleeve indicated the importance of including information related to postharvest treatment, fruit cultivar, and their interaction to understand the effects but failed to incorporate such information in the manuscript.

Section 3.3. one would expect the authors to mention/ include a sentence about the DOI: 10.1080/10408398.2020.1829542 (Biological control of postharvest fungal decays in citrus: a review), at least say the detail about detail of the biocontrol postharvest treatment has been recently covered by Wang et al. (8). Or remove the section since there is no new information

Line 733-735, what are the several edible coatings, these promising results and the type of diseases controlled the author mentioned here. This is a review, and one would need to see this details presented properly. Other wise the authors are just leading the readers to go and check those specific articles referred.

Line 779..’ Vis/NIR techniques applied to fruits suffer from some drawbacks’ please add a reference for this sentence.

Line 780 …..’….fruit or any other sample……’ I thought this is misleading. It should be any biological tissue sample since this idea was taken from aa journal talks about biological tissue and interactions with light.

Line 786. Reference?

Line 807-809 Please consider write this section in detail, what type of citrus fruit and their cultivar included in these studies, what are the external defects you mentioned, list the pathogens ….

Line 820-825 very long sentence

Line 835-837, what are the spectra’s and algorithms Liu et al. (285) used, what analysis, and models presented in this article?

Line 855-857, irrelevant information I suppose, the focus should be citrus here

Line 859-861, what are the type of fruit and vegetables Chen et al. (268) worked on

Line 900, Line 902-904, these studies are on citrus trees, (pre-harvest) , is it relevant here?

Line 907-910, well, this section doesn’t show samples the authors talking about or what type of sensor there is. Please read it.

Line 911-915, this is for example the most important area the authors supposed to address since they are claimed that he aim of this study include early detection. However, at least here there is no information what type of citrus fruit, type of quality monitored or identified diseases. 

After reading the whole manuscript, I still haven’t get the information the authors said will address on the aim of the study ‘with the latest news concerning the early detection of citrus fruit alterations 55 during handling and storage’.

I think it is possible to reduce the number of references if the authors focus on the most important information.

Thank you 

Author Response

Thank you for your suggestions.

An improved version of the manuscript is attached.

Best regards

Reviewer 4 Report

Substantial revision has been preformed. Recommend accepting the manuscript in the present form.

Author Response

Thank you for your response.

A revised version of the file was submitted in accordance with some suggestions from other reviewers.

Best regards